# How Dog Behavior Influences Pet Owner’s Perceptions of Dog Preference for Dental Chews

**DOI:** 10.3390/ani13121964

**Published:** 2023-06-12

**Authors:** Anamarie C. Johnson, Holly C. Miller, Clive D. L. Wynne

**Affiliations:** 1Department of Psychology, Arizona State University, Tempe, AZ 85287, USA; cwynne1@asu.edu; 2General Mills, 1 General Mills Blvd, Golden Valley, MN 55426, USA; holly.miller@genmills.com

**Keywords:** dog, pet food industry, human–dog interaction

## Abstract

**Simple Summary:**

Most studies on dog food and treat preferences focus on owner reports about the product and how much the dog consumes. The aim of this study was to examine dog behavior and engagement in a home-environment with eight different dental chews. Owners submitted a video of their dogs which was analyzed to investigate any relationship between coded dog behavior and owner survey responses for preference among the chew types. Owner-reported dog preference related more to the video coded behavior than their own preference providing some preliminary guidance on what factors might relate to product preference and purchase and how analysis of in-home behavior may better guide pet product research.

**Abstract:**

American pet owners spend billions of dollars on food and treats so it is important to understand what products they want and what they think their dog would enjoy. This study analyzed video recordings of dogs engaging in dental chews in their home environment and compared the observed appetitive behaviors to owner preference and owner-reported dog preference. Overall, appetitive behavior differed significantly between some dental chews. Owner preference for the chews correlated significantly with dog appetitive behavior, but the effect was small (r (702) = 0.22, *p* = 0.001), whereas owner-reported dog preference correlated significantly with dog appetitive behavior and showed a moderate effect size (r (702) = 0.43, *p* = 0.001)—similar in magnitude to findings when parents are asked to report on their children’s behavior. By merging objective behavioral observation of owner-recorded videos with their survey responses, we were able to preliminarily parse out what factors owners may use to assess preference and encourage the future use of in-home video recordings to better understand dog and owner engagement and interaction with pet products.

## 1. Introduction

In 2022, around 69 million households in the United States owned a dog and spent 138.6 billion dollars in the pet industry [1]. Fifty-eight billion dollars were spent on food and treats [1], so understanding what a dog prefers to consume and how that might align with owner preference is economically important. The relationship between owner perceptions of dog preferences and dog behavior also raises basic questions of how well people understand their dogs’ sources of pleasure and discomfort.

The National Research Council defined palatability for dog and cat food as the “physical and chemical properties of the diet which are associated with promoting or suppressing feeding behavior during the pre-absorptive or immediate post-absorptive period” [2] (p. 24). Yet, as argued by Aldrich and colleagues [3], palatability is much more than a substance’s sensory properties and what makes it appealing to a dog, and researchers should seek a more holistic understanding that considers the animal, prior exposure to food, and human and environmental factors.

Within palatability, one can analyze a dog’s mere acceptance of a food product or parse out preference by looking at a deliberate choice for one product over another [3]. Historically, acceptance of food has been analyzed through a single bowl test where the dog is presented with one bowl of food and the amount consumed is measured [3]. By switching between an old and a new food, one can compare acceptance of foods and how intake varies between the two [3]. This method is primarily useful in assessing true dislike of a product; to assess more graded choice, Aldrich and colleagues [3] recommended a two-choice design.

In a two-choice design, a dog is simultaneously given equal amounts of two food products in two different bowls [3] The dog’s preference is assessed by observing the product the dog first approaches, first consumes, and, after a set time, comparing the amount of food remaining in the bowls [3]. This design has the drawback that pet dogs might not be able to discriminate between two items or may be indifferent to them [3]. Laboratory dogs, on the other hand, can be trained to be highly discriminatory, but their refined palates have been shown to be quite different than those of pet dogs [4].

To better understand motivation for a particular food, some modifications of the standard two-choice design have been attempted. Some researchers have added in operant tasks and compared food products using a dog’s motivation to press a lever [5] or perform a certain trained behavior such as touching its nose to the experimenter’s hand [6]. Working within the applied behavior analysis tradition, one study utilized a paired stimulus assessment and a subsequent reinforcer assessment to parse out dog preference [6]. Dogs were offered choices among six food items presented in pairs; the six were then ranked from least to most preferred. Based on those preferences, the least and most preferred food types were later presented pairwise on two different reinforcement schedules where the dogs had to press their noses to the experimenter’s hand to assess preference. The study found that the most preferred food item in the paired stimulus assessment functioned as a reinforcer for both schedule conditions with dogs responding more to the most preferred food item than to the least preferred [6]. Additionally, in a survey, dogs’ owners were usually accurate in predicting the dog’s most preferred food item [6].

Another study merged a two-choice design with direct analysis of dog behavior. Dogs first sampled two food products (a meat treat compared to a bland kibble). Then, during testing, the dogs were prevented from accessing the food by placing it in locked containers. Instead, their behavior in attempting to gain access to the food—such as by pawing at the container or sniffing—was measured and compared [7]. These behaviors on the inaccessible test were then compared to how much of the same products was consumed when they were presented in food puzzles. The authors found that the meatier treat was sniffed and engaged with more often during the inaccessible test than the less-preferred kibble and this preference was maintained during the food puzzle assessment [7].

Beyond direct comparative presentation of food, dog preference is often assessed through owner reports. These studies often relate how the owner sees their dog holistically “enjoying” a food product to their preference for obtaining that product [8,9,10]. Several factors have been shown to influence owner motivation to feed a product. One study analyzed how a pet food’s aroma, appearance, and color might relate to owner preference and owner-reported dog’s preference; owner’s overall liking for a product corresponded with their ratings of its physical qualities, which in turn correlated with owner-reported predicted dog liking [8].

Few studies have considered the specific dog behaviors that might relate to an owner’s assessment of a dog’s preference for one food over another. In a comparison of dogs who ate either a conventional food diet, a raw meat diet, or a vegan diet, owners were asked to report factors that contributed to why they fed the specific food to their dog and whether their dog presented any of the ten palatability behavioral indicators the researchers proposed [11]. These indicators were derived from owner reports from a prior study [12] and included behaviors such as speed of eating, jumping, vocalizing, sniffing the food, and tail wagging [11]. Knight and Satchell [11] determined that certain palatability indicators loaded onto what they identified as dog enthusiasm towards a food (associated with behaviors including tail wagging or eating food quickly). Alternatively, sniffing, as also reported by Di Donfrancesco and colleagues [12], was interpreted as indicating hesitancy in consuming the food [11].

Several studies have investigated dogs’ preferences for different dog foods, but fewer have investigated dogs’ preference and ingestion of items sold as dog treats. White et al. [13] reported that 70% of surveyed dog owners viewed treats as something additional given to their dog, not part of their standard, daily diet. Sixty-two percent of owners identified chews as a common, popular treat and many owners reported giving treats more than three times a week [13]. Owners felt that treats were important to make the dog happy but also noted that treats could be context-specific, such as rewards during training. Owners also indicated that different chews could be given for different reasons—such as dental chews for dental hygienic care [13].

Periodontal disease, which includes both gingivitis and periodontitis, can result from the buildup of plaque on dogs’ teeth which may lead to bacteria that can affect the tissues of the mouth [14]. It has been estimated that up to 85% of adult dogs suffer from some form of periodontal disease and, as a result, many veterinarians recommend regular teeth brushing or veterinary professional teeth cleaning to remove harmful plaque. However, teeth brushing can be difficult for owners to maintain on a daily schedule and putting a dog under anesthetic to carry out teeth cleaning can be risky [14]. As such, giving a daily dental chew that might reduce plaque could be very helpful. In a comparison of three commercially available dental chews, all were shown to inhibit plaque and calculus growth on teeth and reduce halitosis [14].

Offering a dental chew to a dog to facilitate dental care can only be as effective as the dog’s motivation to engage with the chew. Because a dog is unable to articulate its preference for a food, an objective method of assessing preference through behavior has great potential value. No prior study has analyzed how owners’ preferences for different dog treats relate to their reports of their dogs’ preferences and to objective assessment of dogs’ behavior. The aim of this study was to analyze video recordings made by owners during in-home consumer testing and compare the outcome of a direct analysis of the duration and frequency of the dogs’ behaviors to owners’ reports of their own and their dogs’ preferences.

## 2. Materials and Methods

### 2.1. Participants

A market research group with an internal panel of over a million potential participants was contracted to identify, recruit, and perform the consumer testing. Prospective participants were sent a screening questionnaire to determine eligibility based on several criteria relating to the household and owned dogs (Table 1). A target sample size of at least 60 participants was estimated to provide a minimum power of 0.80 to detect a medium effect size at an alpha level of 0.05.

Potential participants were asked to complete an implicit bias test through an online portal to confirm that they owned and utilized a smart phone or internet-connected tablet. Once identified, participants verified that they could commit to the study for a total of four hours over an estimated five weeks. Participants who completed all tasks for the study received $500.

### 2.2. Procedure

Participants were sent the commercially available product bags of eight target dental chews (Naturals Brushless Toothpaste, Arknaturals, Tampa, FL, USA; Blue Dental Bones, Blue Buffalo Company LTD, Joplin, MO, USA; Greenies, Mars Petcare US, Franklin, TN, USA; Merrick Fresh Kisses, Merrick Pet Care, Hereford, TX, USA; Pedigree Dentastix, Mars Petcare US, Franklin, TN, USA; Purina Dentalife, Neehah, WI, USA; Whimzees, Wellness Pet Company, Tewksbury, MA, USA; and Blue Wilderness Wild Bones, Blue Buffalo Company LTD, Joplin, MO, USA). Participants were asked to video film their dogs engaging with each dental chew. To ensure video quality, all participants were first provided written instructions and a sample video and asked to practice filming their dog. Instructions included the owner behaving neutrally towards the dog when the treat was presented and staying out of frame during the filming but present to observe the dog for safety. Ninety participants with the best practice recordings, 30 with small dogs (est. 8–25 lbs; 3.6–11.3 kg), 30 medium dogs (est. 25–45 lbs; 11.3–20.4 kg), and 30 large (45–70 lbs; 20.4–31.8 kg), were selected. All completed practice videos were reviewed by the first author.

Participants with two dogs only submitted responses for one dog and were instructed to keep the other dog separated while the focal dog engaged with the chew.

The study lasted from 9 April 2021 to 14 May 2021.

Each participant presented a single brand of dental chew to their dog for three consecutive days. This same procedure was completed for the remaining seven dental chews brands. The order of presentation of the different brands of dental chew was randomized and assigned across participants.

Participants were instructed to feed a dental chew between meals and to not implement any other dietary changes for their dog during the study duration. On Days 1 and 2, all participants offered the chew to the dog and then completed a series of survey questions relating to the dog’s enjoyment and focus with the chew and its perceived efficacy and freshness. These questions were of interest to the broader scope of the overall project but were not of interest in this study and are not reported here. Participants completed survey questions on Days 1 and 2 and only video recorded their dog on the third day. On Day 3, they were instructed to film the dog from when the chew was offered until it was completely consumed or for at least two minutes. As in the practice filming, owners were asked to behave neutrally and to stay out of frame for the duration of the video. On Day 4, no chew was offered, and participants completed a survey on their preference and perception of the dogs’ preference for that chew. Five questions were selected for analysis; additional questions not relevant to this study are not reported here. All the response options were presented on a Likert scale (Table 2).

### 2.3. Video Analysis

A total of 704 videos, 8 per participant, were analyzed using the event-logging software, BORIS [15], operated by video coders trained to 80% or better accuracy. The video coders were blind to the aims of the study. The first author randomly selected and reviewed 20% of the videos to ensure consistency.

All behaviors were coded for their duration. (Table 3).

Total appetitive engagement was defined from the ethogram as the sum of the four appetitive behaviors (Carry, Chew, Crumbs, and Investigate) from the first 60 s of each video. We selected 60 s for analysis because of a need to standardize a time period during which most dogs were still interacting with the chew (video length varied from 36 to 897 s). Inspection of the video recordings showed that most active engagement with the chews was complete in one minute. Only 2.7% of videos (19 of 704) showed “carry” and 7.7% of videos (54 of 704) showed “investigate” beyond 60 s. There was extensive chewing and interaction with crumbs beyond the first 60 s (85% of videos showed chewing and crumb behavior after 60 s), but we viewed that as a completion of consumption, which depended on the size of the chew and of the dog, rather than evidence of levels of enthusiasm for the chew.

### 2.4. Statistical Analysis

All statistical analyses were conducted using the software package SPSS (Version 28, IBM Corp., Armonk, NY, USA).

A two-way ANOVA was conducted on total appetitive behavior with repeated measures on the type of dental chew and a between-subjects factor of dog weight. Pairwise comparisons with Bonferroni corrections were completed to determine significant differences. Pearson correlation analyses were conducted comparing the five survey questions to each other. Regression analyses were conducted comparing appetitive behavior to the survey questions relating to owner satisfaction in giving the dental chew (happiness or indifference), dog satisfaction in the chew (happiness or disappointment), and owner’s overall liking of the chew.

The frequency of refusal, where the dog did not interact with the chew at any point, for any of the eight dental chews was also calculated as well as how dogs of different weight differed in their refusal of any of the dental chews. Chi square analysis was used to test for differences in brand and dental chew refusal across participants.

## 3. Results

After 5 weeks of the study, 88 participants submitted 704 videos: 28 for small dogs, 31 medium dogs, and 29 large dogs.

A total of 34 dogs (38.6%) refused at least one dental chew: 15 small dogs, 9 medium dogs, and 10 large dogs. There was a total of 116 refusals across 704 chew experiences (16.3%). Purina Dentalife (5.2%) was the least refused chew by all dogs while Whimzees was the most refused (20.9%), (χ^2^ _(2)_ = 13.02, *p* = 0.001).

Behaviors burying, jumping, playing, and vocalizing occurred very infrequently in our sample. Burying was only seen in five participants, jumping in twenty-five participants, playing in six participants, and vocalizing in none.

For the ANOVA on appetitive behavior, Mauchley’s Test of Sphericity indicated that the sphericity assumption was violated (χ^2^ _(2)_ = 59.12, *p* = 0.001) and, thus, a Greenhouse–Geisser correction was used. There was a significant main effect of product (Greenhouse–Geisser corrected, F (5.90, 501.15) = 3.85, *p* = 0.001). Pairwise t-test comparisons with Bonferroni corrections showed significant differences in appetitive behavior between Arknaturals and Dentalife (*p* = 0.01), Dentalife and Merrick (*p* = 0.005), and Greenies and Merrick (*p* = 0.007) (Figure 1). There was also a significant effect of dog weight (F (2, 85) = 3.46, *p* = 0.036). A Tukey’s post hoc analysis showed that large dogs interacted significantly longer with the chews than did small ones (*p* = 0.03). The interaction between appetitive behavior within the first 60 s and dog weight was not significant (Greenhouse–Geisser corrected (F (11.79, 501.15) = 1.37, *p* = 0.180).

Items in the final survey of owners’ preferences and owner-reported dogs’ preferences towards the chews all significantly correlated with each other (Table 4).

We performed a regression analysis on the five survey questions in relation to the coded appetitive behavior (Table 5). For the three questions related to owner preference, the relationships were significant but had a small effect size [16]. Dog behavior significantly predicted owner liking (b = 1.92, t (702) = 6.01, *p* = 0.001, and R^2^ = 0.05), owner happiness in giving the chew (b = 1.59, t (702) = 3.70, *p* = 0.001, and R^2^ = 0.02), and owner indifference in giving the chew (b = 0.95, t (702) = 2.42, *p* = 0.016, and R^2^ = 0.01).

Appetitive behavior accounted for more variance in the two questions relating to owner-reported dog preference, qualifying as moderate effect sizes [16]. Dog behavior significantly predicted owner reporting that the dog liked the chew (b = 4.86, t (702) = 12.55, *p* = 0.001, and R^2^ = 0.18) and owner reporting the dog being disappointed with the dental chew (b = 4.21, t (702) = 12.79, *p* = 0.001, and R^2^ = 0.19).

## 4. Discussion

This study compared direct behavioral analysis of dogs’ engagement with eight different dental chews to owner survey responses. Overall, there were differences in the amount of appetitive behavior in the first 60 s of a dog receiving the dental chews. Owner responses on survey questions relating to their preference and their reports of dog preference for a chew significantly correlated with each other. Coded appetitive behavior correlated significantly with owner preference and with owner-reported dog preference, but the correlations differed in magnitude: owner preference had only a small relationship with dog behavior, whereas the effect sizes of the relationship between owner-perceived dog preferences and dog behavior were medium [16].

Some palatability-related behaviors frequently reported in the literature were not displayed by the dogs in our sample. Jumping and vocalizing, reported as indicators of palatability and enthusiasm towards food by Knight and Satchell [11], occurred infrequently (jumping) or not at all (vocalizing) in our sample. Knight and Satchell’s [11] study, however, relied solely on owner reports of behavior rather than objective coding. As the authors noted, behavior assessments by untrained owners are less reliable than analysis of video recordings by trained observers blind to the aims of the study. By comparing owner-reported dog preference to objective behavioral coding, participants in our sample did appear to appropriately use some of their dog’s behavior to report their dog’s preference.

Prato-Previde and colleagues [17] showed that owners can have a large impact on how dogs interact with food. When owners displayed interest in a bowl containing a few pieces of kibble compared to a bowl with more, dogs counterintuitively chose the bowl with less food. When there was no owner influence, dogs chose the bowl with more kibble [17]. We attempted to minimize owner influence by instructing owners not to interact with their dog while it was engaging with the chew during filming, but we were only able to view the interaction from one camera angle and did not have recording of the first two dental chew engagements. It is, thus, possible that the behavior we observed might have been influenced by the owner. For example, the instruction to remain neutral may have led to more subdued behavior in the dogs which could explain why we did not observe much jumping or vocalizing as reported by Knight and Satchell [11].

Total appetitive behavior coded from video recordings accounted for about 18% of the variance in owner responses about their dog’s preferences for the dog chews. Similar trends of correlations of moderate magnitude are seen in studies that have compared parental reports to direct analysis of children’s behavior [18]. Stifter and colleagues [18] noted that several studies found significant correlations between parent ratings and unbiased observations but that these correlations were often weak to moderate in magnitude (r < 0.30). A study by Root and Stifter [19] compared mothers’ questionnaire responses of how they would react to their child’s negative emotions to unbiased observations in laboratory and classroom settings. Mothers’ reported behavioral responses of support correlated with the observed laboratory and classroom behavior, accounting for a moderate effect size, R^2^ = 0.14.

It is interesting that, although dog behavior predicted a moderate amount of the variance in owners’ reports of dogs’ perception of dental chews, it predicted very little of owners’ reports of their own preference for the chews. This could indicate that other dog behaviors not coded here, or unanalyzed aspects of the chew, such as its size, color, or branding, may influence owner responses. It is also possible that when asked about their own preference among chew brands, people focus solely on their own impressions of texture, odor, and possible flavor, independent of anything their dog is doing. In a comparison of parent and child food preferences, researchers found that while there was overall similarity in preference, parents would often focus their preferences on the healthiness of a food which was not a factor in the children’s own preferences [20]. Similarly, when parents were asked to report their preference for children’s books and then predict their child’s preference, parents preferred books with cultural acclaim while factors such as whether their child’s gender matched the book’s protagonist or the number of words per page affected how they predicted their child’s preference [21].

The evidence here that participants were able to competently record and upload video recordings of their dogs’ behavior opens the door for objective behavioral observation of dogs in their home without the intervention of strangers and, thus, where they are most comfortable. With some simple instructions, participants in our study were able to record footage that was then analyzed by trained coders and compared to previous studies that had relied on owner reporting of their dog’s behavior which may be subject to bias and misinterpretation [11,12]. By merging objective observation with survey data, we gained insight into what features owners might be using when making a purchasing decision. Future research should expand on the opportunity that in-home video recording provides to obtain an understanding of how dogs and owners interact with pet food products.

### Limitations

One possible limitation of this study was the variety of durations with which the dogs interacted with the chews and, consequently, the varied video-recording durations provided by the owners. We attempted to control for this by only analyzing the first 60 s of each video and found that some appetitive behaviors such as “investigate” and “carry” occurred at relatively low frequencies after that time. Chewing persisted beyond 60 s, but most dogs did not chew for more than around 40 of the first 60 s. Interaction with crumbs also persisted beyond 60 s but crumb behavior could only occur if the chew was complete and soft enough to form crumbs. However, the nature of a dog’s interaction with a dental chew is not necessarily just a function of the dog’s enthusiasm for it, but also related to its size, hardness, and abrasiveness.

Additionally, while participants were instructed to feed their dog between meals so their dog would not be satiated, we could not know the feeding schedule for dogs and how that might affect chew interest.

It might have been beneficial to have owners record their dogs’ behavior with the dental chew over all three days of chew exposure. This would have made it possible to observe possible changes in behavior as the dogs adjusted to the chews. Dogs tend to be more neophilic than other animals and prefer variety in their daily diet [22]. Callon et al. [23] and Vondran [24] noted that dogs showed enhanced interest in novel food products. However, dogs can also present neophobic behavior and, as Callon and colleagues [23] noted, dogs presented a novel food showed more hesitation and were slower to eat on the first day of food presentation than the last. Thus, it seems likely that behavioral interaction with the chews we presented would have changed over the course of three days, however, neophilic or neophobic behavior was not the focus of this study.

## 5. Conclusions

This study utilized owner-reported surveys and owner-recorded videos during an in-home test of eight commercially available dental chews to investigate how dogs’ overt behavior corresponded to owner survey responses. Overall durations of appetitive engagement during the first 60 s across the different brands of dental chew were quite similar. We found the observed dogs’ behavior only had a small impact on owner preference while the same behavior predicted owner-reported dog preference with a moderate effect size. Researchers should take advantage of in-home video recording to better understand how owners perceive their dog’s liking of a product and how that ultimately may affect their preference and purchasing intent.

## Figures and Tables

**Figure 1 animals-13-01964-f001:**
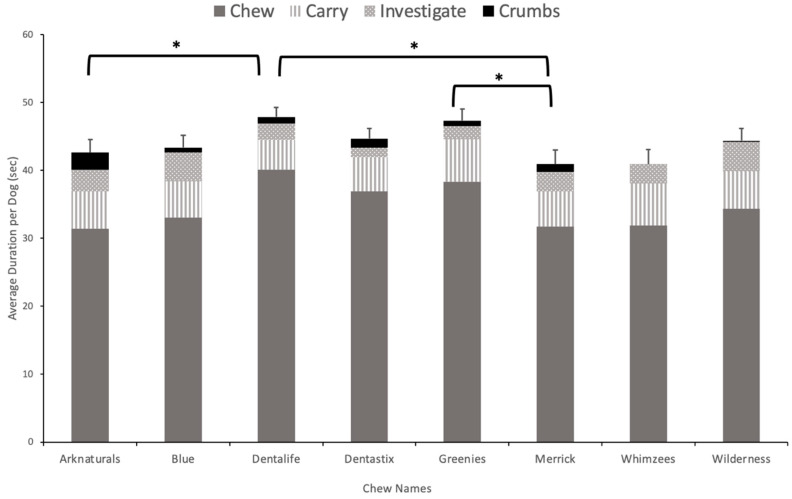
Comparison of appetitive behaviors of dogs across the eight dental chews in first 60 s of exposure. Error bars represent standard error. * indicates significant difference in total interaction, *p* < 0.05.

**Table 1 animals-13-01964-t001:** List of criteria for inclusion in the survey.

Name of Criterion	Criterion Eligibility
Age	25–54 years old
Household Composition	Exclude grown children living with parents
Household Income	Greater than $30,000
Level of Education	Minimum high school graduate
Employment	Full- or part-time employed, retired, homemaker
Occupation type	NOT employed in advertising, market research, sales promotion, media, veterinarian or animal hospital, pet store, manufacturer pet supplies, or new product development
Number of Dogs in home	One or two dogs
Relationship to Dog in home	Make all or most purchase decisions for dog, schedule, and attend more veterinarian visits
Age of dog	1–10 years old
Muzzle shape	Medium to very long snout, no brachycephalic dogs
Last visit to veterinarian	Within last 6–8 months
Treat purchase and offering to dog	Regularly purchase and offer dental chews daily to once a week, need to have purchased a dental chew within the last 4 months
Type of Dental Chew purchase	One of eight brands in study: Arknaturals, Blue Dental Bones, Purina Dentalife, Pedigree Dentastix, Greenies, Merrick Fresh Kisses, Whimzees, or Blue Wilderness Wild Bones
Medical history of dog	No history of digestive issues after giving treats or new food, no dental issues, no food allergies
Technology	iPhone or Android, download of survey application

**Table 2 animals-13-01964-t002:** List of selected five questions from Day 4 survey with scale labels where applicable.

Overall, how well do you like or dislike these dog dental chews?	1: Dislike Extremely 2: Dislike very much 3: Dislike moderately 4: Dislike slightly 5: Neither like nor dislike 6: Like slightly 7: Like moderately 8: Like very much 9: Like extremely
I felt happy giving these dental dog chews to my dog	1: Not at all happy 2: 3: 4: Neither happy or unhappy 5: 6: 7: Very happy
I felt indifferent when giving these dental dog chews to my dog	1: Very Indifferent 2: 3: 4: Neutral 5: 6: 7: Not at all indifferent
My dog felt happy when I offered him/her these dental dog chews	1: Dog felt not at all happy 2: 3: 4: Neutral 5: 6: 7: Dog felt very happy
My dog felt disappointed when I offered him/her these dog dental chews	1: Very disappointed 2: 3: 4: Neutral 5: 6: 7: Not at all disappointed

**Table 3 animals-13-01964-t003:** Ethogram of coded behaviors.

Coded Behavior	Definition
Chew	Dog is actively eating/ingesting the chew.
Ignore	Dog does not interact with the chew. Dog’s head is turned away from chew or dog is near chew but is not smelling/engaging with it.
Carry	Dog is holding the chew in its teeth but is either actively moving/walking with the chew while not chewing or it has the chew in its mouth and is holding it without eating/ingesting.
Investigate	Dog is not chewing but smelling/investigating the chew. Behaviors can include sniffing, pawing, or licking the chew.
Hold	Dog chews by holding chew between two paws or using one of its paws to prop the chew up.
Crumbs	Chew is complete but dog licks or seeks out remaining crumbs.
Jumping	Dog jumping up for chew, at least two paws off the ground at the same time.
Dancing	Dog is tapping paws in fast movement while waiting to receive chew.
Burying	Dog attempts to bury the chew either in outside substrate or in furniture or bedding, presented behavior can be moving head back and forth.
Playing	Dog tosses chew around, play bows around chew.

**Table 4 animals-13-01964-t004:** Descriptive statistics (number (n), mean (M), and standard deviation (SD)) of Likert Scale (1–7) survey responses and their Pearson intercorrelation matrix. Questions relating to indifference and disappointment were reverse coded.

Questions	n	M	SD	1	2	3	4	5
1. Overall liking of the chew	704	6.70	1.99	--				
2. Owner felt happy with the chew	704	5.37	1.49	0.79 **	--			
3. Owner felt indifferent about the chew	704	5.17	1.65	0.50 **	0.59 **^a^	--		
4. Dog felt happy about the chew	704	5.67	1.51	0.62 **	0.61 **	0.43 **^a^	--	
5. Dog felt disappointed about the chew	704	5.74	1.77	0.57 **	0.53 **^a^	0.32 **	0.80 **^a^	--

** Indicates significance level of *p* < 0.01 (2 tailed). ^a^ Indicates questions were reverse coded.

**Table 5 animals-13-01964-t005:** Regression analysis for survey questions on dental chews with the predictor variable appetitive behavior.

	R	R Square	Adj.R Square	Unstandardized B	*p*
Overall liking	0.22	0.05	0.05	1.92	0.001
Human felt happy	0.14	0.02	0.02	1.59	0.001
Human felt indifferent	0.09	0.01	0.01	0.95	0.016
Dog felt happy	0.43	0.18	0.18	4.86	0.001
Dog felt disappointed	0.44	0.19	0.19	4.21	0.001

## Data Availability

The data presented in this study are available on request from the corresponding author. The data are not publicly available due to data pertaining to private industry.

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
