# Peer review of "How Dog Behavior Influences Pet Owner’s Perceptions of Dog Preference for Dental Chews"

_animals, 2023, doi:10.3390/ani13121964_

Round 1

Reviewer 1 Report

Line 31 needs citation.

I think it would help underscore the importance of the study if you highlighted the fact that owner's were appropriately reading their dogs behaviors and were understanding the canine behaviors that indicated a preference for the specific chew.  Other studies have suggested that people were not able to correctly interpret canine behaviors as to their meaning re: behavioral indications of agression for example.  So this appears to be the major takeaway from the article and it is buried in the lead.  It is also not suppo0rted in the literature review as its not mentioned at all.  

Author Response

Thank you so much for your feedback. Please find notes below where edits were made in the manuscript per your comments. Comments are also highlighted in the manuscript for location

Line 31 needs citation.

   Citation was added, information was from Reference 1

I think it would help underscore the importance of the study if you highlighted the fact that owner's were appropriately reading their dogs behaviors and were understanding the canine behaviors that indicated a preference for the specific chew.  Other studies have suggested that people were not able to correctly interpret canine behaviors as to their meaning re: behavioral indications of agression for example.  So this appears to be the major takeaway from the article and it is buried in the lead.  It is also not supported in the literature review as its not mentioned at all.  

Text added: "By comparing owner-reported dog preference to objective behavioral coding, participants in our sample did appear to appropriately use some of their dog’s behavior to report their dog’s preference."

Reviewer 2 Report

The use of dental chews in feeding dogs affects their dental health, but their interest in them depends on several factors (one of which is palatability). The authors analysed the behaviour of dogs in the context of their preferences in this area and those of their owners. This manuscript is rarely addressed in the scientific literature and appears attractive from a canine welfare perspective.

The paper's title reflects its scope, and the aim is clearly stated. The methodology is adequate, the description of the results is well done, and the inference is correct.

My only doubt is whether: "...palatability is much more than a substance's sensory properties..." (lines 39-40). Palatability depends on sensory properties. Perhaps the authors meant "attractiveness"?

Author Response

Thank you so much for your feedback. Please find notes below where edits were made in the manuscript per your comments. Comments are also highlighted in the manuscript for location

My only doubt is whether: "...palatability is much more than a substance's sensory properties..." (lines 39-40). Palatability depends on sensory properties. Perhaps the authors meant "attractiveness"?

            Text was updated to read: “Yet, as argued by Aldrich and colleagues [3], palatability is much more than a substance’s sensory properties and what makes it appealing to a dog, and researchers should seek a more holistic understanding that considers the animal, prior exposure to food, and human and environmental factors.”

Reviewer 3 Report

The study looks at dog preference for dental treats, and how this correlates with owner reported preference, and owner reported dog preference of those treats. The inclusion of correlation with owner reported preferences made the study interesting. The paper is well-written, but I do have some concerns with some aspects of it. There are some parts of the methods that are unclear (specified below), and while the discussion that is there is ok, what I am missing is the conclusions that the authors draw from their research. Please see my detailed comments below.

Abstract

Generally the abstract needs to follow the same format as the paper; intro, methods, results, discussion. Obviously that all needs to be shortened to fit the maximum allowable number of words. However, this abstract only appears to contain methods and results, it is missing a (very short) intro and discussion.

Methods

Line 157. The remaining participants were asked to take photos instead of video. It is a little unclear why this was the case were their recordings of too low quality to ensure correct analysis? If so (or for other reason), it wil help the reader understand better if the reason is explicitly stated here.

Line 160. Did the participants present a random dental chew to their dog out of the 8 they received? How did they pick which one to give their dog on those 3 days?

Line 169. dog preference for a chew brand. This is unclear, how can the participants compare chew brands if they have only given the single brand of dental chew to their dog for 3 days? Please clarify.

Line 174/175. Ok, it is explained a bit more here. Please consider moving this description up in the methods (around line 160), so the reader gets an easier understanding of the experiment.

How were the photos analysed from the participants who were asked to take photos instead of video? The authors state that they plit the participants up in 90 taking videos and the remainder taking photos, however, they do not state how they analyse the photo data. It does not seem to be included in this research at all. This needs to be explained in the methods, or the authors clearly need to state that only the video formed part of the analysis. Also, how does a group of 90 videoing participants meet the target of 180 that the authors state (earlier in the methods) they require for robustness of statistical analysis?

It would have been interesting to find out why owners reported preferences, and give them space to provide a reason for their answer of choice to each question.

Results

Ok that 209 participants completed all tasks, but why is this mentioned if the data from only 88 participants is analysed? What happened to the photo data?

Table 4 needs more of a description. Tables need to be able to be understood as a stand-alone item; from the description of Table 4, the reader still does not understand what is being presented. M = mean, ok, but mean of what? What tests were done that these are the results of? Try to make it as easy as possible for the reader to understand what they are looking at. I am assuming these are the results from the correlation analysis between appetetive behaviour and survey responses, but if not, than the methods section needs to be updated with the description of the analysis the authors did here.

Discussion

The discussion is fine, but what I am missing at the end is some form discussion about what the results of this study mean in the broader context of dog food preference, and the implications for further research. The conclusions that are presented at the end of the discussion are simply a summary of the findings of the study. However, I think by that point the reader understands what the findings of the study are already. What do the results mean for our understanding of dog food preferences (or in this case dental treats)? What does this mean in the broader context of dog food preference research? Given the results that the authors found, what are their conclusions from this research (as opposed to what are the results)?

This is missing from the abstract as well.

Author Response

The study looks at dog preference for dental treats, and how this correlates with owner reported preference, and owner reported dog preference of those treats. The inclusion of correlation with owner reported preferences made the study interesting. The paper is well-written, but I do have some concerns with some aspects of it. There are some parts of the methods that are unclear (specified below), and while the discussion that is there is ok, what I am missing is the conclusions that the authors draw from their research. Please see my detailed comments below.

Abstract- max 200 words

Generally the abstract needs to follow the same format as the paper; intro, methods, results, discussion. Obviously that all needs to be shortened to fit the maximum allowable number of words. However, this abstract only appears to contain methods and results, it is missing a (very short) intro and discussion.

Updated abstract:

American pet owners spend billions of dollars on food and treats so it is important to understand what products they want and what they think their dog would enjoy. This study analyzed video recordings of dogs engaging in dental chews in their home environment and compared the observed appetitive behaviors to owner preference and owner-reported dog preference. Overall, appetitive behavior differed significantly between some dental chews. Owner preference for the chews correlated significantly with dog appetitive behavior, but the effect was small (r (702) = .22, p = .001), whereas owner-reported dog preference correlated significantly with dog appetitive behavior and showed a moderate effect size (r (702) = .43, p = .001) — similar in magnitude to findings when parents are asked to report on their children’s behavior. By merging objective behavioral observation of owner-recorded video with their survey responses, we were able to preliminarily parse out what factors owners may use to assess preference.

Methods

Line 157. The remaining participants were asked to take photos instead of video. It is a little unclear why this was the case – were their recordings of too low quality to ensure correct analysis? If so (or for other reason), it wil help the reader understand better if the reason is explicitly stated here. 

Any text reference to the alternative group was removed for simplicity

  • Line 160. Did the participants present a random dental chew to their dog out of the 8 they received? How did they pick which one to give their dog on those 3 days?

Text Lines 174 on added to beginning of the paragraph

  • Line 169. ‘dog preference for a chew brand’. This is unclear, how can the participants compare chew brands if they have only given the single brand of dental chew to their dog for 3 days? Please clarify.

Text updated to read and clarified: “On Day 4, no chew was offered, and participants completed a survey on their preference and perception of the dogs’ preference for that chew’

  • Line 174/175. Ok, it is explained a bit more here. Please consider moving this description up in the methods (around line 160), so the reader gets an easier understanding of the experiment.

See update to Line 160 comment

  • How were the photos analysed from the participants who were asked to take photos instead of video? The authors state that they split the participants up in 90 taking videos and the remainder taking photos, however, they do not state how they analyse the photo data. It does not seem to be included in this research at all. This needs to be explained in the methods, or the authors clearly need to state that only the video formed part of the analysis. Also, how does a group of 90 videoing participants meet the target of 180 that the authors state (earlier in the methods) they require for robustness of statistical analysis?

Any text reference to the alternative group was removed for simplicity

  • It would have been interesting to find out why owners reported preferences, and give them space to provide a reason for their answer of choice to each question.

Results

  • Ok that 209 participants completed all tasks, but why is this mentioned if the data from only 88 participants is analysed? What happened to the photo data?

Any text reference to the alternative group was removed for simplicity

  • Table 4 needs more of a description. Tables need to be able to be understood as a stand-alone item; from the description of Table 4, the reader still does not understand what is being presented. M = mean, ok, but mean of what? What tests were done that these are the results of? Try to make it as easy as possible for the reader to understand what they are looking at. I am assuming these are the results from the correlation analysis between appetetive behaviour and survey responses, but if not, than the methods section needs to be updated with the description of the analysis the authors did here.

Table updated to describe the mean: Descriptive statistics (number (n), mean (M), and standard deviation, (SD)) of Likert Scale (1-7) identified survey responses and their Pearson intercorrelation matrix. Ques-tions relating to indifference and disappointment were reverse coded.

Text under 2.4 Statistical analysis added the line to highlight the specific pearson correlation analysis reflected in Table 4: Pearson correlation analyses were conducted comparing the five identified survey questions to each other.

Discussion

  • The discussion is fine, but what I am missing at the end is some form discussion about what the results of this study mean in the broader context of dog food preference, and the implications for further research. The conclusions that are presented at the end of the discussion are simply a summary of the findings of the study. However, I think by that point the reader understands what the findings of the study are already. What do the results mean for our understanding of dog food preferences (or in this case dental treats)? What does this mean in the broader context of dog food preference research? Given the results that the authors found, what are their conclusions from this research (as opposed to what are the results)? This is missing from the abstract as well.

Text added to discussion:

The evidence here that participants were able to competently record, and upload video recordings of their dogs’ behavior opens the door for objective behavioral observation of dogs in their home without the intervention of strangers and thus where they are most comfortable. With some simple instructions, participants in our study were able to record footage that was then analyzed by trained coders and compared to previous studies that had relied on owner reporting of their dog’s behavior which may be subject to bias and misinterpretation [11,12]. By merging objective observation with survey data, we gained insight into what features owners might be using when making a purchasing decision. Future research should expand on the opportunity that in-home video recording provides to get an understanding of how dogs and owners interact with pet food products.

Conclusion text updated:

This study utilized owner-reported surveys and owner-recorded video during an in-home test of eight commercially available dental chews to investigate how dogs’ overt behavior corresponded to owner survey responses. Overall durations of appetitive engagement during the first 60 seconds across the different brands of dental chew were quite similar. We found the observed dogs’ behavior only had a small impact on owner preference while the same behavior predicted owner-reported dog preference with a moderate effect size. Researchers should take advantage of in-home video recording to better understand how owners perceive their dog’s liking of a product and how that ultimately may affect their preference and purchasing intent.

Reviewer 4 Report

This was a very interesting study and one I had not read about previously. I'm not an expert in statistical analysis so cannot comment on that part, but the design seems reasonable and well thought out.   

I have two very minor suggestions:

Line 97 - use "and" rather than ampersand for reference (Knight and Satchell)

Line 163 - just provide a very brief explanation of why irrelevant questions were asked. 

Author Response

Thank you so much for your feedback. Please find notes below where edits were made in the manuscript per your comments. Comments are also highlighted in the manuscript for location

I have two very minor suggestions:

Line 97 - use "and" rather than ampersand for reference (Knight and Satchell)

   Text was updated to replace & with “and”

Line 163 - just provide a very brief explanation of why irrelevant questions were asked. 

Text added: “These questions were of interest to the broader scope of the overall project but were not of interest of this study and are not reported here”

Reviewer 5 Report

This is a well-written manuscript, with a good Introduction that adequately places the research into context. Authors provide a good review of the scientific background of their study and highlight the relevance of the paper.

Similarly, the Discussion and the Limitations section are well constructed and do not overinterpret the findings. Where I found some additional clarifications would be necessary, these were the Methods and Results sections.

Methods

It is not clear, how many of each of the 8 target chews were provided to the participants. It seems like owners had to feed 3 consecutive days with one type of the chews, then on day 4, complete a survey, then continue with the next type of chew. Is this correct?

Were there any special instructions, which part of the day the experimental chews should be given to the dogs? Were the owners instructed, how to present the chews? Acting like this would be something extra yummie? Or just behave normally? These details could affect some of the behaviors later the Authors list in the ethogram table (anticipatory behaviors). Was it specified, whether the chews should be given to the dogs after a given amount of time since the last meal the dog ate?

Did the authors instruct the owners NOT to feed another type of dental chew in the regime of the testing?

Results

When showing the results of chew-refusal, it would be interesting to know, whether these dogs were just fed before the chew was offered to them.

In case of the appetitive behavior durations, after the ANOVA provided a significant main effect, why the Authors did not use Tukey post hoc test when looking for significant between-group differences?

Author Response

Thank you so much for your feedback. Please find notes below where edits were made in the manuscript per your comments. Comments are also highlighted in the manuscript for location

Methods

It is not clear, how many of each of the 8 target chews were provided to the participants. It seems like owners had to feed 3 consecutive days with one type of the chews, then on day 4, complete a survey, then continue with the next type of chew. Is this correct?

Text added: “This same procedure was completed for the next seven dental chews, each completed over a 4-day period.”

Were there any special instructions, which part of the day the experimental chews should be given to the dogs? Were the owners instructed, how to present the chews? Acting like this would be something extra yummie? Or just behave normally? These details could affect some of the behaviors later the Authors list in the ethogram table (anticipatory behaviors). Was it specified, whether the chews should be given to the dogs after a given amount of time since the last meal the dog ate?

Text added: “Instructions included the owner behaving neutrally to the dog when the treat was pre-sented and staying out of frame during the filming but present to observe the dog for safety.”

    There are additional comments in the discussion that note that this neutral behavior from the owner could have subdued the dog’s normal behavior

“Participants were instructed to feed a dental chew between meals and to not implement any other dietary changes for their dog during the study duration”

Did the authors instruct the owners NOT to feed another type of dental chew in the regime of the testing?

Results

When showing the results of chew-refusal, it would be interesting to know, whether these dogs were just fed before the chew was offered to them. 

Text added in discussion: “While participants were instructed to feed their dog between meals so their dog would not be satiated, we could not know the feeding schedule for dogs and how that might affect chew interest”

In case of the appetitive behavior durations, after the ANOVA provided a significant main effect, why the Authors did not use Tukey post hoc test when looking for significant between-group differences?

Pairwise t test were used because we were looking at within subject comparisons for product length of appetitive behavior across the chews

Round 2

Reviewer 3 Report

It was nice seeing this paper again, and the authors did a great job incorporating my comments from the previous review. I do think some little things would benefit from a bit more tweaking, and when the authors removed all reference to the participants who did not video their dogs, some other important information was also lost in the methods. Detailed comments below.

Simple summary. From the instructions to authors fromt eh journal, the simple summary should include: a clear statement of the problem addressed, the aims and objectives, pertinent results, conclusions from the study and how they will be valuable to society. The results and conclusions from this study are missing from the simple abstract.
It should also be written for a lay audience, without any technical terms without explanation. I am not sure  a lay audience would know what holistic behavioural analysis is.

Abstract. The last line/conclusion in the abstract does not match the concluding message from the discussion. Is the most important message from the discussion that the factors that affect owner preference were preliminary identified (as stated in the abstract), or that in-home video recording is a good tool (discussion)? What is the most important message that follows the results from this research? That message should be included at the end of the abstract, and should really match the take-home message from the conclusion/discussion.

Methods

Line 150-151. To ensure their dog. This sentence is a little surprising for the reader here, as it has not been mentioned previously that the researchers asked the participants to take videos. It would be worthwile to mention that method first, before describing it in more detail.

Line 156. video group. I understand that this is phrased this way because there was another non-video group. However, all reference to the non-video group have been removed, therefore, calling this the video group is very confusing.

Line 161. each participant days This would be more clear if phrased as each participant presented a single brand of dental chew to their dog for three consecutive days.

Line 162. next seven dental chews. This is unclear. Maybe better to state remaining seven brands of dental chews.

Line 161-162. a four day period. This needs more clarification here, as the chews are only presented for 3 days. It needs clarification where the 4th day comes in to all this.

Line 166-172. Given the detailed description of day 1, 2 and 4, it leaves the reader wondering what happened to day 3? It would help this section of text if a sentence was also added about day 3.

Line 178-179. this same . Video analysis. Is that supposed to be here?

Line 203. pearson . other. Remove identified.

Results

Line 213 214. after dogs. This sentence is missing the number for medium dogs, and there seems to be a single closing parenthesis with a comma there, which I suspect is not supposed to be there?

Line 218-219. While . Interest. This is really a sentence for the discussion, not the results.

Line 230-232. There was . small ones. This is a little confusing here. This section deals with the results from the appetitive behaviour analysis, which only spans the first 60 seconds of video. Why, in the middle of this do the authors present results from a different analysis (total length of interaction)? It makes more sense to present all the results from the appatitive analysis together in one spot, and then (or before that) present results from different analysis.

Table 4 description. Remove identified in front of survey responses. I am unsure what identified means here, or what it refers to.

Line 237 Remove identified in front of survey questions.  I am unsure what identified means here, or what it refers to.

Discussion

Line 275. did not have recordings of the first two dental chew engagements. This is not described in the methods, the methods give the impression that all dental chew engagements were recorded. If only the third and last engagement with each brand of dental chew was recorded, that should be specified in the methods. Otherwise, this sentence needs to be rephrased for clarity.

Lines 305-315 and lines 338-348 are exactly the same. The authors needs to pick the location for this section that they find most appropriate and delete the other part.

Author Response

It was nice seeing this paper again, and the authors did a great job incorporating my comments from the previous review. I do think some little things would benefit from a bit more tweaking, and when the authors removed all reference to the participants who did not video their dogs, some other important information was also lost in the methods. Detailed comments below.

Simple summary. From the instructions to authors fromt eh journal, the simple summary should include: a clear statement of the problem addressed, the aims and objectives, pertinent results, conclusions from the study and how they will be valuable to society. The results and conclusions from this study are missing from the simple abstract. 
It should also be ‘written for a lay audience, without any technical terms without explanation’. I am not sure  a lay audience would know what ‘holistic behavioural analysis’ is.

Abstract. The last line/conclusion in the abstract does not match the concluding message from the discussion. Is the most important message from the discussion that the factors that affect owner preference were preliminary identified (as stated in the abstract), or that in-home video recording is a good tool (discussion)? What is the most important message that follows the results from this research? That message should be included at the end of the abstract, and should really match the take-home message from the conclusion/discussion.

Text for simple summary and abstract updated

 Simple Summary: Most studies on dog food and treat preferences focus on owner reports about the product and how much the dog consumes. The aim of this study was to examine dog behavior and engagement in a home-environment with eight different dental chews. Owners submitted video of their dogs which was analyzed to investigate any relationship between coded dog behavior and owner survey responses for preference among the chew types. Owner-reported dog preference related more to the video coded behavior than their own preference providing some preliminary guidance on what factors might relate to product preference and purchase and how analysis of in-home behavior may better guide pet product research.

Abstract: American pet owners spend billions of dollars on food and treats so it is important to understand what products they want and what they think their dog would enjoy. This study analyzed video recordings of dogs engaging in dental chews in their home environment and com-pared the observed appetitive behaviors to owner preference and owner-reported dog preference. Overall, appetitive behavior differed significantly between some dental chews. Owner preference for the chews correlated significantly with dog appetitive behavior, but the effect was small (r (702) = .22, p = .001), whereas owner-reported dog preference correlated significantly with dog appetitive behavior and showed a moderate effect size (r (702) = .43, p = .001) — similar in magnitude to findings when parents are asked to report on their children’s behavior. By merging objective behavioral observation of owner-recorded video with their survey responses, we were able to preliminarily parse out what factors owners may use to assess preference and encourage the future use of in-home video recordings to better understand dog and owner engagement and interaction with pet products.

Methods

Line 150-151. ‘To ensure … their dog’. This sentence is a little surprising for the reader here, as it has not been mentioned previously that the researchers asked the participants to take videos. It would be worthwile to mention that method first, before describing it in more detail.

Text added:

Participants were asked to film their dogs engaging with each dental chew.

Line 156. ‘video group’. I understand that this is phrased this way because there was another non-video group. However, all reference to the non-video group have been removed, therefore, calling this the video group is very confusing.

Text of video group removed

Sentence now reads:

Ninety participants with the best practice recordings, 30 with small dogs (est. 8 – 25lbs; 3.6 - 11.3kg), 30 medium dogs (est. 25 - 45lbs; 11.3 - 20.4kg), and 30 large (45 – 70lbs; 20.4 - 31.8kg), were selected.

Line 161. ‘each participant … days’ This would be more clear if phrased as ‘each participant presented a single brand of dental chew to their dog for three consecutive days’.

Text added:

Ninety participants with the best practice recordings, 30 with small dogs (est. 8 – 25lbs; 3.6 - 11.3kg), 30 medium dogs (est. 25 - 45lbs; 11.3 - 20.4kg), and 30 large (45 – 70lbs; 20.4 - 31.8kg), were selected.

Line 162. ‘next seven dental chews’. This is unclear. Maybe better to state ‘remaining seven brands of dental chews’.

Line 161-162. ‘a four day period’. This needs more clarification here, as the chews are only presented for 3 days. It needs clarification where the 4th day comes in to all this.

Text added per the two comments above:

This same procedure was completed for the remaining seven dental chews brands, each completed over a 3-day period

Line 166-172. Given the detailed description of day 1, 2 and 4, it leaves the reader wondering what happened to day 3? It would help this section of text if a sentence was also added about day 3.

Text added:

On Day 3, they were instructed to film the dog from when the chew was offered until it was completely consumed or for at least two minutes. As in the practice filming, owners were asked to behave neutrally and to stay out of frame for the duration of the video.

Line 178-179. ‘this same …. Video analysis’. Is that supposed to be here?

Unsure what this reviews to. I see “this same” in sentence 160 and then the section of 2.3 Video Analysis

Line 203. ‘pearson …. other’. Remove ‘identified’.

“Identified: was removed

Results

Line 213 – 214. ‘after … dogs’. This sentence is missing the number for medium dogs, and there seems to be a single closing parenthesis with a comma there, which I suspect is not supposed to be there?

Text edited and updated:

28 for small dogs, 31 medium dogs, and 29 large dogs

Line 218-219. ‘While …. Interest’. This is really a sentence for the discussion, not the results.

Text moved to 4.1 Limitations

Line 230-232. ‘There was …. small ones’. This is a little confusing here. This section deals with the results from the appetitive behaviour analysis, which only spans the first 60 seconds of video. Why, in the middle of this do the authors present results from a different analysis (total length of interaction)? It makes more sense to present all the results from the appatitive analysis together in one spot, and then (or before that) present results from different analysis.

Clarifying text added: all analysis in this paragraph was on appetitive behavior

There was also a significant effect of dog weight on appetitive behavior (F (2, 85) = 3.46, p = .036).

Table 4 description. Remove ‘identified’ in front of ‘survey responses’. I am unsure what ‘identified’ means here, or what it refers to.

Updated Table with “identified” removed

Line 237’ Remove ‘identified’ in front of ‘survey questions.  I am unsure what ‘identified’ means here, or what it refers to.

“identified” removed

Discussion

Line 275. ‘did not have recordings of the first two dental chew engagements’. This is not described in the methods, the methods give the impression that all dental chew engagements were recorded. If only the third and last engagement with each brand of dental chew was recorded, that should be specified in the methods. Otherwise, this sentence needs to be rephrased for clarity.

Text added:

Participants completed survey questions on Days 1 and 2 and only video recorded their dog on Day 3.

Lines 305-315 and lines 338-348 are exactly the same. The authors needs to pick the location for this section that they find most appropriate and delete the other part.

Text paragraph to be at end of discussion, not limitations.